# Effects of Water Extract from *Artemisia argyi* Leaves on LPS-Induced Mastitis in Mice

**DOI:** 10.3390/ani12070907

**Published:** 2022-04-01

**Authors:** Qianbo Ma, Yuanhao Wei, Zitong Meng, Yuhua Chen, Guoqi Zhao

**Affiliations:** Department of Pratacultural Science, College of Animal Science and Technology, Yangzhou University, Yangzhou 225009, China; mqb1601555436@163.com (Q.M.); weiyuanhao1911@163.com (Y.W.); zulutango7@163.com (Z.M.); mz120201324@yzu.edu.cn (Y.C.)

**Keywords:** *Artemisia argyi*, lipopolysaccharide, mastitis, oxidative stress

## Abstract

**Simple Summary:**

Mastitis is a common disease in dairy cows. On the one hand, it will reduce milk yield and milk quality of dairy cows, thus increasing the cost of animal husbandry, and, on the other hand, it will influence the health of infected animals and even human beings. Generally speaking, because mastitis is caused by pathogenic microorganisms, antibiotic treatment is commonly used. However, antibiotic resistance of microorganisms caused by wrongful use of antibiotics and antibiotic residues after treatment has become an urgent problem to be solved. Chinese herbal medicines are pure natural substances, and many of them have antibacterial, anti-inflammatory, or immune-enhancing effects. In this experiment, *Artemisia argyi* (*A. argyi*) was selected as the research object to construct the cell model of cow mastitis. Studies have found that *A. argyi* extract can play a positive role in the regulation of inflammation, which is rich in organic acids and flavonoids. Therefore, *A. argyi* extract may be a potential treatment for mastitis.

**Abstract:**

In the context of the unsatisfactory therapeutic effect of antibiotics, the natural products of plants have become a research hotspot. *Artemisia argyi* (*A. argyi*) is known as a traditional medicine in China, and its extracts have been reported to have a variety of active functions, including anti-inflammatory. Therefore, after establishing the mouse mastitis model by lipopolysaccharide (LPS), the effects of *A. argyi* leaves extract (ALE) were evaluated by pathological morphology of the mammary gland tissue, gene expression, and serum oxidation index. Studies have shown that ALE has a restorative effect on LPS-induced mammary gland lesions and significantly down-regulated the rise of myeloperoxidase (MPO) induced by LPS stimulation. In addition, ALE played a positive role in LPS-induced oxidative imbalance by restoring the activities of glutathione peroxidase (GSH-PX) and superoxide dismutase (SOD) and preventing the increase in nitric oxide (NO) concentration caused by the over-activation of total nitric oxide synthase (T-NOS). Further analysis of gene expression in the mammary gland showed that ALE significantly down-regulated LPS-induced up-regulation of inflammatory factors IL6, TNFα, and IL1β. ALE also regulated the expression of MyD88, a key gene for toll-like receptors (TLRs) signaling, which, in turn, regulated TLR2 and TLR4. The effect of ALE on iNOS expression was similar to the effect of T-NOS activity and NO content, which also played a positive role. The IκB gene is closely related to the NF-κB signaling pathway, and ALE was found to significantly alleviate the LPS-induced increase in IκB. All of these results indicated that ALE may be considered a potential active substance for mastitis.

## 1. Introduction

Mastitis is a common disease that affects bovine production. It directly affects lactation ability and milk quality, which has become the focus of attention. Mastitis is generally caused by bacterial infections. Studies have shown that the main pathogens are *Escherichia coli* (*E. coli*) and *Staphylococcus aureus* (*S. aureus*) [1]. According to different bacterial pathogens, mastitis is divided into two types: contagious mastitis and environmental mastitis. Clinical mastitis in bovines is often caused by *E. coli*, and it causes significant production losses and animal welfare problems for dairy farms worldwide and can lead to the death of the animal [2]. Antibiotics remain the most widely accepted treatment for mastitis worldwide because of their low cost and ease of use [3]. In order to avoid poor antibiotic treatment, excessive antibiotics are often used to take into account the infection caused by various pathogens. The above problems inevitably lead to the excessive and incorrect use of antibiotics, leading to the resistance of pathogenic bacteria and the problem of antibiotic residues. Studies have shown that no new antibiotics have come to market since 1984, suggesting that pathogenic bacteria have had enough time to adapt to them and develop resistance [4]. *E. coli* has a special position in the field of microbiology, generally as symbionts with mutual reciprocity and mutual benefit with the host, but studies have found that *E. coli* exists in strains resistant to all antibiotics. This is because of the particularity of the internal structure of *E. coli*, meaning it can not only produce its own drug resistance genes but also combine foreign drug resistance genes or even pass the drug resistance genes to other bacteria [5].

From the point of view of health, the residues caused by the abuse of antibiotics have caused concern. In the context of the problem of anti-resistance and drug resistance of pathogenic bacteria, the treatment of mastitis has gradually become diversified [6,7,8]. Due to the pure nature, low toxicity, and low residue, plant components are not easy to produce drug resistance, so studies have been carried out in order to replace antibiotics. In China, there is a long history of medicinal plants. Many studies have shown that Chinese herbal medicine has the effects of anti-inflammation, regulating the internal environment, improving immunity, and so on. *Artemisia argyi* (*A.argyi*) is a traditional Chinese herbal medicine, which has been proven to be rich in flavonoids, polysaccharides, and volatile oil [9]. In addition, different extracts of *A. argyi* leaves have been reported to have antioxidant, antibacterial, and immunomodulatory effects [10,11,12].

Lipopolysaccharide (LPS) is the cell wall component of gram-negative bacteria *E. coli*. A large number of studies have shown that LPS released by gram-negative bacteria is an important stimulating factor of inflammation in bovine mammary epithelial cells (bMECs) [13,14]. The occurrence of infection in the body is often accompanied by the generation of superoxide free radicals, and it has been proved that oxidative stress is closely related to inflammation [15]. Oxidative stress causes excessive production of reactive oxygen species (ROS), and antioxidant enzymes play an important role in balancing ROS [16,17]. Studies have shown that oxidative stress-induced imbalances include decreased levels of glutathione peroxidase (GSH-PX) and superoxide dismutase (SOD), and increased malondialdehyde (MDA) and myeloperoxidase (MPO) [18]. In previous studies, we found that *A. argyi* leaves water extract (ALE) positively regulates LPS-induced inflammation in bMECs [19]. At present, researchers, through the establishment of a mouse mastitis model, simulate the cause of bovine mastitis and then explore the pathogenesis and treatment methods [20,21].

Therefore, in order to further explore the role of ALE in vivo, the mouse mastitis model was selected for study. The purpose of this study was to use the established LPS-induced mouse mastitis model to explore the mechanisms of ALE from morphology, pathology, oxidative stress, and the expression of related genes.

## 2. Materials and Methods

### 2.1. Preparation of A. argyi Leaves Extract

*A. argyi* was harvested in Suqian, Jiangsu, China. The leaves of *A. argyi* were placed in an open area away from the sun to dry naturally. Liposoluble substances and part pigment in crushing *A. argyi* leaves were removed with petroleum ether with a boiling range of 60–90 °C. Then, *A. argyi* leaves were extracted with 95% ethanol after removing petroleum ether from them. Next, the combined and concentrated extract was extracted again with water three times, and the system was separated with a high-speed centrifuge to obtain the supernatant. Finally, the *A. argyi* leaves extract’s (ALE) dry powder was prepared by a freeze-drying mechanism.

### 2.2. Animals

Forty SPF BALB/c female mice, aged 8–10 weeks (20.0 ± 0.5 g), were purchased from the Animal Experiment Center of Yangzhou University within 3 days after parturition. The purchased mice were divided into 4 groups with 2 cages in each group and 5 mice in each cage. The mice were placed in a clean environment to strictly ensure animal welfare. Three days before the experiment, mice were allowed to eat and drink freely to adapt to the new environment, and the mastitis model was established on the fourth day.

### 2.3. Experimental Design

After 3 days of adaptive feeding, we first performed a clinical examination of the mice before the experiment to confirm their healthy status. Then, the mice were intraperitoneally injected with 50 μL 10% chloral hydrate (TCA). After anesthesia, 4 groups were treated according to Table 1 below, in which LPS (*E. coli* O55:B5) was purchased from Sigma (Ronkonkoma, NY, USA). The liquids in Table 1 were dissolved in a phosphate buffer solution (PBS) and filtered by a 0.22 μm membrane before use. At the beginning, the control check (CK) and ALE groups were injected with 50 μL sterile PBS, and the LPS and LPS + ALE groups were injected with 50 μL LPS into the two nipples of the fourth pair. A total of 12 h later, the CK group was injected with 50 μL sterilized PBS, the LPS group was injected with 50 μL LPS, the ALE group was injected with 50 μL ALE, and the ALE + LPS group was injected with 50 μL ALE containing LPS.

### 2.4. Histopathologic Evaluation

After 24 h of treatment, the mice were euthanized. The skin of the mice was cut along the midabdominal line until the fourth lacteal region was exposed. First, the breast area was photographed for visual observation. After the photo was taken, some tissues were collected for histopathological examination. Some tissue was placed in 4% paraformaldehyde, embedded in paraffin, and sliced. After hematoxylin and eosin (HE) staining, the pathological changes in the mammary tissue were assessed under a light microscope (×100), and CellSens Dimension software (Olympus Corporation, Tokyo, Japan) was used for section photography.

### 2.5. Antioxidant Evaluation

After 24 h of treatment, blood was taken from orbit to obtain mouse serum. The activities of SOD, GSH-PX, and T-NOS and the content of NO in the serum were determined by SOD, GSH-PX, T-NOS, and NO kits. All kits were purchased from Nanjing Jiancheng Institute of Biological Engineering, and the specific determination method refers to the kit instructions.

### 2.6. Inflammatory Infiltration Evaluation

Part of the collected mammary tissue was taken. Each sample was accurately weighed, and MPO activity was determined by an MPO kit (Nanjing Jiancheng, Nanjing, China). The specific determination method refers to the kit instructions.

### 2.7. RNA Isolation and RT-qPCR

Part of the collected mammary tissue (30 mg) was placed in 1.5 mL of an enzyme-free fingerless tube, and 1 mL of lysate (Trizol) and small steel balls were also added to the tube. Homogenates were formed through a tissue grinding apparatus. A total RNA extraction kit was used to isolate RNA from the homogenates (TIANGEN, Shanghai, China). The real-time PCR kit was used to synthesize the cDNA and was subject to PCR amplification. NCBI was used to query gene sequences, and Primer Premier 6.0 was used to design the primers. The information on these primers is listed in Table 2. The real-time PCR cycle conditions were as follows: an initial denaturation at 95 °C for 30 s, followed by 40 cycles of 95 °C for 5 s and annealing at 60 °C for 30 s. GAPDH was used as a reference gene, and expression levels were calculated using 2^−ΔΔCT^.

### 2.8. Statistical Analysis

Before data analysis, Kolmogorov—Smirnov and Levene tests were used to study the normal distribution and homogeneity of variance. Then, the mean value ± SEM was used to represent statistical analysis. SPSS 22.0 software (IBM, Armok, New York, USA) was used for the independent sample T-test. Images were made by GraphPad Prism 6 software (GraphPad Software, California, USA), and the results in *p* < 0.05 or *p* < 0.01 were considered statistically significant.

## 3. Results

### 3.1. Effects of ALE on Histopathological Changes

#### 3.1.1. Morphological Observation of Mammary Gland in Mice

LPS action in the breast usually causes clinical mastitis. In this study, we first treated mice for 12 h and observed with naked eyes that LPS treatment caused redness and swelling in the mammary area, while the other groups showed normal behavior. After an extra 12 h of different treatments, the mammary gland tissue was observed by autopsy. As shown in Figure 1, sterile PBS was injected into the control group, and the mammary gland of mice showed light flesh color, with clear peripheral blood vessels and a healthy growth state. Compared with the control group, the mammary gland in the LPS group showed obvious changes. The color of the mammary gland was deepened, showing deep silt purple, and showed hyperplasia, attached to the blood vessels, while the color of blood vessels was deepened and thickened, and the symptoms of hyperemia and swelling appeared. The ALE group showed similar mammary tissue morphology to the control group. In the LPS + ALE group, darker mammary tissue was also observed, but the color was lighter than in the LPS group, and the symptoms of vascular congestion and swelling were reduced. Based on the above results, the LPS inflammation model was successfully constructed, so the influence of ALE on the inflammation model can be evaluated by further experiments.

#### 3.1.2. Pathological Observation of the Mammary Gland in Mice

As shown in Figure 2, in the control group, the acinar wall of the breast tissue section was clearly visible, and the structure was intact without damage, with almost no inflammatory cell infiltration. Compared with the control group, the pathological changes in the LPS group were obvious. The acinar wall of the mammary gland showed an unclear state, while the cell wall was thickened, and there was a form of structural destruction. More noticeable was inflammatory cell infiltration, presenting purple granules and distributing around the cells. This result also proved the success of the mouse mastitis model constructed by the LPS infusion of milk ducts. The ALE group had little difference compared with the control group; there was a small amount of inflammatory infiltration, but the acinar structure was intact, and the boundary was clear. Compared with the LPS group, the infiltration degree of inflammatory cells was lower and the structure of the acinar wall was more intact in the ALE + LPS group.

#### 3.1.3. Inflammatory Infiltration of Mammary Tissue

MPO activity reflects the level of neutrophils, which can reflect the degree of inflammation in the mammary tissue. The results are shown in Figure 3. The addition of LPS significantly increased the activity of MPO (*p* < 0.01). The up-regulation of MPO by ALE was significantly lower than that by LPS (*p* < 0.05). In addition, ALE significantly reduced the LPS-induced increase in MPO activity (*p* < 0.01).

### 3.2. Serum Oxidative Stress Indexes

By separating the serum from the whole blood, the activities of SOD, GSH-PX, and T-NOS and the content of NO in the serum were further determined. The results in Figure 4 showed that LPS-induced inflammation significantly decreased SOD and GSH-PX activity (*p* < 0.01) and significantly increased the activity of T-NOS (*p* < 0.01). When ALE was added, SOD activity increased significantly (*p* < 0.01) but had no significant effect on GSH-PX and T-NOS activity (*p* > 0.05), and there was only an upward trend in numerical value. It was found that ALE significantly up-regulated LPS-induced activity reduction in SOD and GSH-PX (*p* < 0.01). Positive effects of T-NOS were also observed, and ALE significantly mitigated the effects of LPS (*p* < 0.01). The change of NO content was consistent with that of T-NOS. Although the addition of ALE also significantly increased NO content. It significantly down-regulated the increase in NO content caused by LPS addition (*p* < 0.05).

### 3.3. Mammary Tissue Related Genes Expression

This study further investigated the effect of LPS on breast tissue and the regulatory effect of ALE on LPS by measuring the expression of related genes in mammary tissue. The expression levels of three inflammatory factors, three toll-like receptor-related genes, one NO synthesis related gene, and one NFκB activation related gene were determined. As shown in Figure 5, the expression levels of IL6, TNFα, and IL1β in breast tissues were significantly increased by the addition of LPS (*p* < 0.01). Although the addition of ALE also significantly up-regulated their expression levels (*p* < 0.01), the changes were negligible compared with the addition of LPS. Meanwhile, the addition of ALE significantly down-regulated LPS-induced overexpression levels (*p* < 0.01). LPS significantly increased the expression levels of MyD88, TLR2, and TLR4 (*p* < 0.01), but ALE supplementation significantly down-regulated TLR4 expression (*p* < 0.01) and had no significant effect on the expression of MyD88 and TLR2 (*p* > 0.05). ALE also significantly down-regulated toll-like receptor expression induced by LPS supplementation (*p* < 0.01), especially the expression of TLR4, which returned to the normal level. LPS also affected the synthesis of NO and significantly increased the expression of the iNOS gene (*p* < 0.01). The addition of ALE significantly down-regulated iNOS expression and LPS induced high expression (*p* < 0.01). LPS significantly up-regulated the expression of IκB, a key gene in the NFκB pathway (*p* < 0.05). ALE also significantly up-regulated the expression of IκB (*p* < 0.01) but significantly down-regulated LPS-induced high expression (*p* < 0.01).

## 4. Discussion

Mastitis, as a conventional clinical disease of bovines, is usually caused by pathogenic microorganisms. It not only affects the welfare of bovines but also seriously hinders the development of the dairy industry. Studies have shown that infections caused by *E. coli* are considered the most common cause of mastitis death [22]. LPS is a unique structure in the cell wall of gram-negative bacteria, including *E. coli*, which binding protein (LBP) induces further inflammatory responses by binding to the LPS receptor (CD14) [23]. In previous studies, ALE has been found to regulate LPS-induced bovine mammary epithelial cells. Therefore, in order to further explore the mechanism of ALE, LPS was used to construct a mouse inflammatory model, observe the pathological changes of the mammary gland in mice, and evaluate the role of ALE in vivo.

In this experiment, the inflammation model of mice was induced, and the fourth milk region of mice was selected for induction. On the one hand, the growth of this milk region was more developed than the other four milk regions of mice, and, on the other hand, studies showed that this milk region was similar to the mammary gland of cows [24]. Although, many studies have shown differences between mouse and cow mammary glands. Therefore, mouse mastitis model studies should be evaluated in conjunction with bovines. The results also showed a high degree of similarity between mouse and cow results, suggesting that the advantages of mice can be used to screen for potential antibacterial and immunomodulatory compounds. The key problem is that potential drugs may not initially be available in sufficient quantity to be tested in bovines, and mice would be able to solve this problem.

Pathological changes in tissue are an indicator of inflammatory response. Polymorphonuclear neutrophils (PMNS) are the most important natural defense mechanism for mastitis, accompanied by white blood cells metastasis to the mammary gland [25,26]. In this experiment, adding LPS to breast tissue caused severe tissue pathological changes. After staining with hematoxylin and eosin, the acinar wall was thickened, and there was a wide range of inflammatory cells, such as neutrophils and macrophages, in the acinar lumen, and some acinar walls were injured, which was consistent with most of the damage presented by the LPS-induced mouse mastitis model [27,28]. The addition of ALE has a significant alleviating effect on LPS-induced injury. Many studies have found that many natural substances can reduce LPS-induced mouse mammary tissue damage, such as baicalein [29], leonurine [30], and sophocarpine [31], which means it is possible to use natural products to replace antibiotics.

In addition to tissue staining, changes in MPO can also reflect the severity of inflammation, and higher MPO content means stronger inflammatory injury [20] because MPO activity is a sign of neutrophil influx into tissue and an important indicator for evaluating tissue injury [32]. In this study, LPS significantly increased the activity of MPO, while the addition of ALE significantly reduced the negative effects of LPS, which was consistent with the results of pathological changes.

Studies have shown that the antioxidant system is an effective sensor linking oxidative stress and inflammation [33]. When the body is stimulated by LPS, the levels of antioxidant enzymes reflect the body’s ability to resist oxidative stress [15]. In this study, the addition of LPS caused inflammatory damage to breast tissue, resulting in a decrease in GSH-PX and SOD levels. The addition of ALE significantly up-regulated SOD. Although the up-regulation of GSH-PX was not significant, ALE showed a recovery effect on the down-regulation of SOD and GSH-PX induced by LPS. NO has a wide range of biological effects, plays an important role in immune response and inflammatory regulation, and is a major source of ROS [34]. Therefore, the excessive activation of T-NOS activity leads to the excessive increase in NO, which will lead to the imbalance of ROS. The LPS-induced elevation in this study was mitigated by the addition of ALE. ALE has shown a positive effect on oxidative stress caused by inflammation.

In order to further explore the mechanism of LPS, the expression of related genes in mouse mammary tissue was determined. LPS has been found to activate the NF-κB signaling pathway, which leads to the activation of many inflammatory factors, including IL1β, IL6, TNFα, etc. [35,36]. It was consistent with the results of this study, and the addition of ALE alleviated the LPS-induced upregulation. The expression of IκB, a key gene in NF-κB activation, was significantly increased by LPS to activate NF-κB, and ALE also had a positive effect on IκB. MyD88 is an important connector protein and a key gene of toll-like receptors (TLRs) [37]. TLRs are ligands for the innate immune system to recognize pathogens, and many studies have shown that TLR4 recognizes LPS [38,39]. In this study, LPS induced the upregulation of MyD88, which further led to the upregulation of TLR2 and TLR4. ALE had a particularly significant effect on TLR4, which was significantly down-regulated compared with the control, and finally brought the effects of LPS to a normal level. ALE also showed positive effects on LPS induced MyD88 and TLR2, similar to the expression of related genes in bMECs [19]. Finally, the expression of the iNOS gene was also determined, which was closely related to the production of NO, consistent with the trend of T-NOS activity and NO content in mouse mammary tissue.

## 5. Conclusions

In this study, we first established a mouse mastitis model by LPS and further found the alleviating effect of ALE on LPS-induced mastitis. It not only directly alleviated the tissue damage of mice from pathological morphology but also had a good recovery effect on the oxidative stress caused by LPS. Meanwhile, it also had a positive effect on inflammatory transmission and oxidative regulation genes in breast tissue. Therefore, ALE may be a promising bioactive extract for alleviating the negative effects of LPS.

## Figures and Tables

**Figure 1 animals-12-00907-f001:**
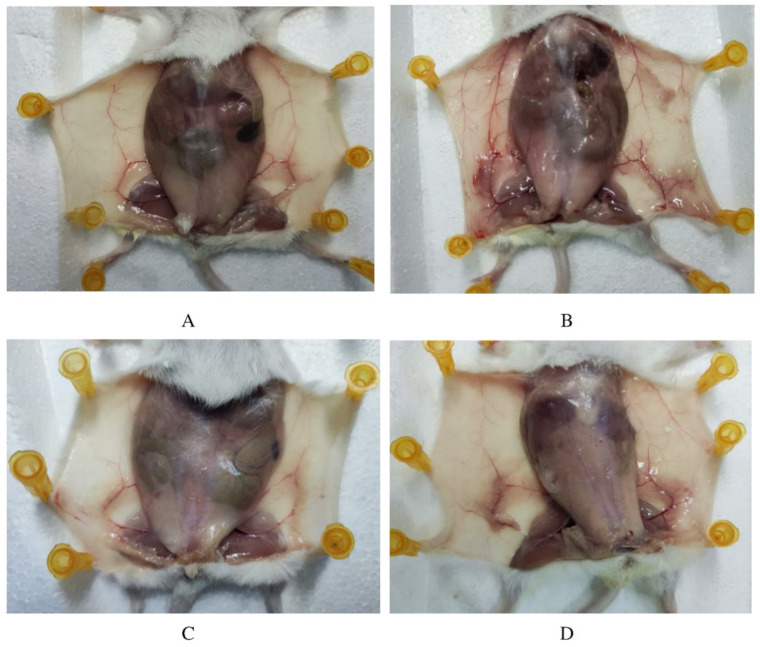
ALE alleviated LPS-induced mammary inflammation in mice. Note: (**A**): Control group, (**B**): LPS group, (**C**): ALE group, (**D**): LPS + ALE group.

**Figure 2 animals-12-00907-f002:**
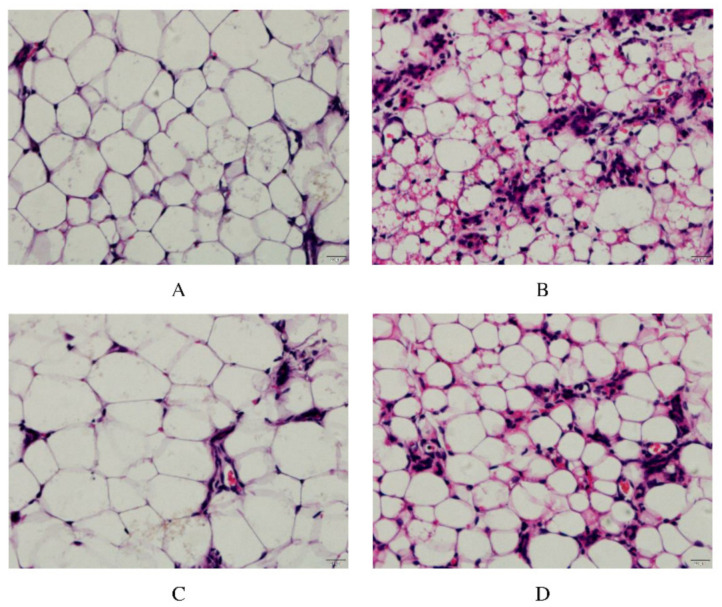
ALE alleviated pathological damage of the LPS-stimulated mammary gland in mice. Note: (**A**): Control group, (**B**): LPS group, (**C**): ALE group, (**D**): LPS + ALE group.

**Figure 3 animals-12-00907-f003:**
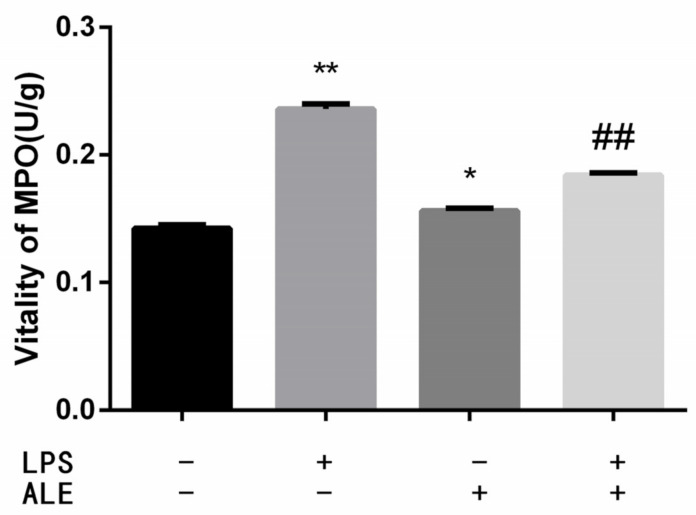
ALE reduced MPO activity in LPS-induced mouse mastitis. * *p* < 0.05, ** *p* < 0.01 vs. control group; ## *p* < 0.01 vs. LPS group.

**Figure 4 animals-12-00907-f004:**
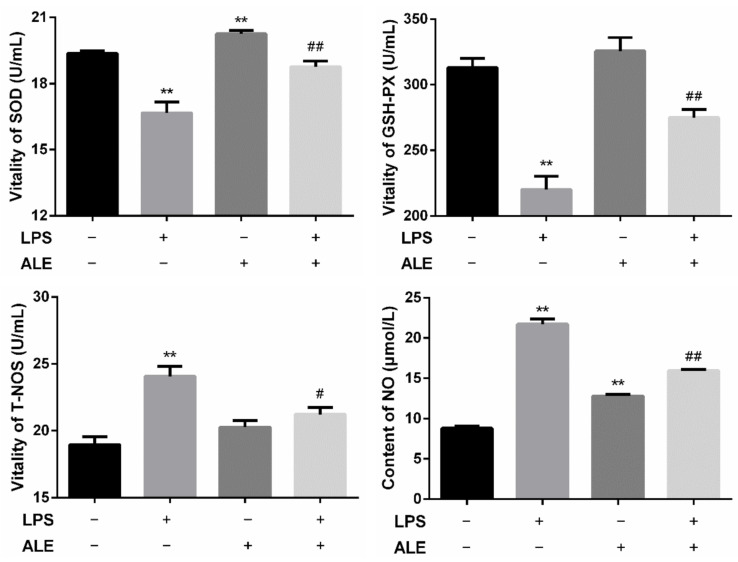
Regulation of ALE on serum oxidation indexes in LPS-induced mice. ** *p* < 0.01 vs. control group; # *p* < 0.05, ## *p* < 0.01 vs. LPS group.

**Figure 5 animals-12-00907-f005:**
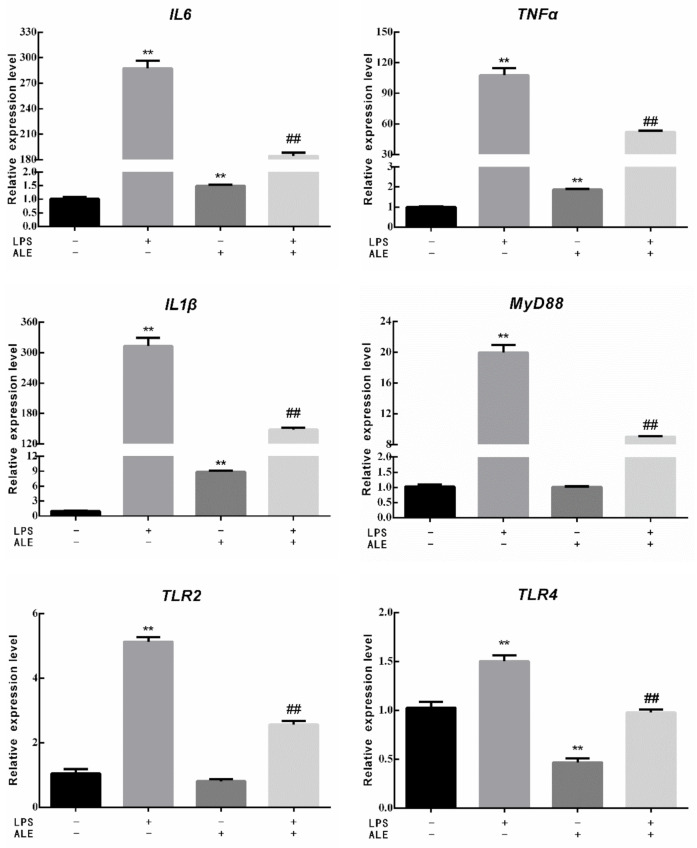
Effects of ALE on LPS-induced mammary tissue related genes in mice. * *p* < 0.05, ** *p* < 0.01 vs. control group; ## *p* < 0.01 vs. LPS group.

**Table 1 animals-12-00907-t001:** Design of inflammatory induction in mice.

Times	Groups			
	CK	LPS	ALE	LPS + ALE
0 h	PBS	0.2 mg/mL LPS	PBS	0.2 mg/mL LPS
12 h	PBS	0.2 mg/mL LPS	10 mg/mL ALE	10 mg/mL ALE with 0.2 mg/mL LPS

**Table 2 animals-12-00907-t002:** Primers used in this study.

Gene	Primer Sequence ^1^, 5′-3′	Accession No.	Size
GAPDH	F: TCTCCTGCGACTTCAACA	NM_001289726.1	117
	R: TGTAGCCGTATTCATTGTCA		
IL6	F: TCCATCCAGTTGCCTTCT	NM_001314054.1	137
	R: TAAGCCTCCGACTTGTGA		
TLR2	F: TGGAGGTGTTGGATGTTAG	NM_011905.3	253
	R: GATAGGAGTTCGCAGGAG		
TLR4	F: TTCACCTCTGCCTTCACT	NM_021297.3	224
	R: GGACTTCTCAACCTTCTCAA		
TNFα	F: GTGGAACTGGCAGAAGAG	NM_013693.3	278
	R: GCTACAGGCTTGTCACTC		
iNOS	F: CAGGAGATGTTGAACTATGTC	NM_010927.4	272
	R: TTGGTGTTGAAGGCGTAG		
IL1β	F: CTTCAGGCAGGCAGTATC	XM_006498795.5	166
	R: CAGCAGGTTATCATCATCATC		
MyD88	F: CCGTGAGGATATACTGAAGG	NM_010851.3	279
	R: TTAAGCCGATAGTCTGTCTG		
IκB	F: CCTCAGATACCTACCTCACT	NM_010908.5	125
	R: TAGCCTCCAGTCTTCATCA		

^1^ F: Forward primer; R: Reverse primer.

## Data Availability

The data presented in this study are available on request from the corresponding author.

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
