# Peer review of "Effects of Water Extract from Artemisia argyi Leaves on LPS-Induced Mastitis in Mice"

_animals, 2022, doi:10.3390/ani12070907_

Round 1
Reviewer 1 Report
With the help of evidence-based experimental biology, the prospects for the development of medicinal preparations from plants that have long been used in traditional Chinese medicine are substantiated in the work, proposed for publication . The problem raised is undoubtedly of interest. However, there are several questions and comments on the work done.
- At the very beginning of the article, the abbreviation ALE is used (line 23), the decoding in the text is given much further (line 81). I believe that a decoding should be given already at the first use of this abbreviation.
- Line 137-138: “Total RNA extraction kit was used to isolate proteins from homogenates…”. Perhaps they meant "to isolate RNA"?
- Line 158: "LPS infection of the breast usually causes clinical mastitis." I consider this proposal incorrect, since the term "infection" is applicable to pathogens - bacteria, viruses, etc.
- Lines 223-225: “As shown in Figure 5, the expression levels of IL6, TNFα and IL1β in breast tissues were significantly increased by the addition of LPS (p < 0.01). Addition of ALE also significantly increased their expression levels (p < 0.01)…” indeed ALE increase the expression of these cytokines, but dozens of times less prominent compared to LPS, so it is incorrect to write “significantly increased” in both cases.
- And further lines 225-226 “…but ALE significantly down-regulated the expression level of LPS (p < 0.01)” Perhaps they meant “expression level of IL6, TNFα and IL1β on the background of LPS stimulation”?
- Why were non-lactating animals used in the development of a mouse model of mastitis?
- Why were such high concentrations of ALE (10 mg/mL) used in the work? In their other work, to which the authors refer, it was shown that ALE at a concentration of 1 μg/ml has the strongest cytotoxic effect, cell survival is less than 20%.
- LPS concentration is also unreasonably high (0.2 mg/mL). Mouse models of mastitis have previously been described in which LPS was administered at a concentration of 20 ng/μL (Xiao HB, Wang CR, Liu ZK, Wang JY. LPS induces pro-inflammatory response in mastitis mice and mammary epithelial cells: Possible involvement of NF-κB signaling and OPN. Pathol Biol (Paris). 2015 Feb;63(1):11-6. doi: 10.1016/j.patbio.2014.10.005. Epub 2014 Nov 4. PMID: 25468491.)
- Why do the authors believe that substances that suppress the expression of inflammatory mediators are a good alternative to antibiotics for mastitis? Inflammation is the immune system's way of fighting infectious agents. If plant extracts, along with an anti-inflammatory effect, also have a bacteriostatic or bactericidal effect, then this must be demonstrated in experiments.
Reviewer 2 Report
I appreciate the idea of the manuscript. However, I recommend introducing some changes as follows:
Line 21: Remove one "." after hotspot
Please define the abbreviation ALE in the abstract
Line 42-43- I don`t understand the division based of "former" and "latter". Please clarify
Line 91 Preparation ?
Table 1 please provide explanation for CK, ALE etc. The table shoud stand alone.
Line 117- maybe change "kill" for "euthanised"
Lines 153- 155 - delete
Line 248 - space befor citation
Lines 328 -331 - have you used bovines in the study ?
lines 333-334 - correct "material"
Reviewer 3 Report
see attached file

Round 2
Reviewer 3 Report
see attached file
